# *SocraticLM*: Exploring Socratic Personalized Teaching with Large Language Models

**Jiayu Liu**[1,2]    **Zhenya Huang**[1,2*]  **Tong Xiao**[1,2]   **Jing Sha**[2]    **Jinze Wu**[2]
**Qi Liu**[1,2]    **Shijin Wang**[2]    **Enhong Chen**[1,2*]

1: University of Science and Technology of China
2: State Key Laboratory of Cognitive Intelligence
{jy251198,tongxiao2002}@mail.ustc.edu.cn;
{huangzhy,qiliuql,cheneh}@ustc.edu.cn;
{jingsha,jzwu4,sjwang3}@ifytek.com

## Abstract

Large language models (LLMs) are considered a crucial technology for advancing intelligent education since they exhibit the potential for an in-depth understanding of teaching scenarios and providing students with personalized guidance. Nonetheless, current LLM-based application in personalized teaching predominantly follows a "Question-Answering" paradigm, where students are *passively* provided with answers and explanations. In this paper, we propose *SocraticLM*, which achieves a Socratic "Thought-Provoking" teaching paradigm that fulfills the role of a real classroom teacher in *actively* engaging students in the thought process required for genuine problem-solving mastery. To build *SocraticLM*, we first propose a novel *"Dean-Teacher-Student"* multi-agent pipeline to construct a new dataset, *SocraTeach*, which contains 35K meticulously crafted Socratic-style multi-round (equivalent to 208K single-round) teaching dialogues grounded in fundamental mathematical problems. Our dataset simulates authentic teaching scenarios, interacting with six representative types of simulated students with different cognitive states, and strengthening four crucial teaching abilities. *SocraticLM* is then fine-tuned on *SocraTeach* with three strategies balancing its teaching and reasoning abilities. Moreover, we contribute a comprehensive evaluation system encompassing five pedagogical dimensions for assessing the teaching quality of LLMs. Extensive experiments verify that *SocraticLM* achieves significant improvements in the teaching performance, outperforming GPT4 by more than 12%. Our dataset and code is available at `https://github.com/Ljyustc/SocraticLM`.

## 1  Introduction

Large language models (LLMs) have achieved impressive results across a variety of tasks including natural language processing, translation, and question-answering [54, 56, 59]. This draws widespread attention to the potential of using LLMs to revolutionize intelligent education, especially personalized teaching [2, 23, 39], mainly due to their two advantages. On one hand, LLMs have displayed human-like mastery and proficiency of knowledge in fundamental subjects like math and physics [1, 48]. Therefore, they can effectively deliver subject-specific instructions to students like a real teacher. On the other hand, LLMs exhibit exceptional understanding and adaptability to users' (i.e., students') inputs. They are capable of comprehending diverse students' demands, providing instant feedback, and engaging in interactive pedagogy. Based on these two aspects, there has been a lot of research

---

*Corresponding Authors

38th Conference on Neural Information Processing Systems (NeurIPS 2024).

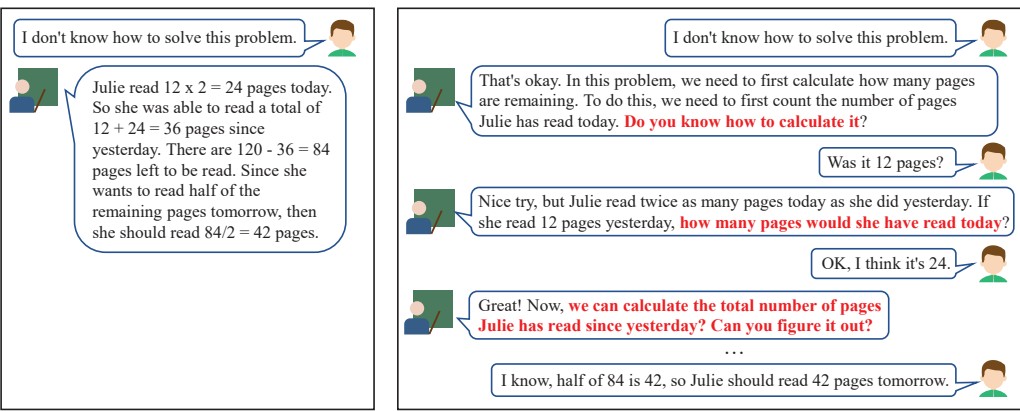

(a) "Question-Answering" teaching paradigm    (b) "Thought-Provoking" teaching paradigm

Figure 1: Teaching paradigms: "*Question-Answering*" vs "*Thought-Provoking*".

investigating to apply a general LLM (e.g., ChatGPT) to personalized teaching [5, 18, 44, 51], or building specific teaching LLMs, such as MathGPT [2], EduGPT [3], and EduChat [10].

However, current LLMs-based personalized teaching methods predominantly adhere to a "*Question-Answering*" paradigm. As shown in Figure 1(a), they passively offer functionalities such as providing answers to questions and explaining knowledge concepts to students' queries. In this process, they oversimplify the teaching into a series of Q&As, directly delivering complete answers based on CoT [56], ToT [57], etc., which falls short of truly identifying the issues students may have and offering targeted assistance. Consequently, students may struggle to comprehend the problem-solving process, lack a genuine improvement in their ability, and fail to resolve similar issues in the future.

In this paper, we draw inspiration from the Socratic method of teaching [13, 45] and propose *SocraticLM*, which achieves a novel "*Thought-Provoking*" teaching paradigm as depicted in Figure 1(b). The key of this paradigm is to engage students in a dialogue to active participation in the learning process, which continually poses open-ended questions (marked red, e.g., "... how to calculate it?") to encourage them to articulate their thoughts, challenge assumptions, and think independently. This process enables students to learn to solve a problem by themselves, thereby fostering a deeper mastery and ability. Compared with LLM-based applications using prompt engineering directly (e.g., GPT4), we aim to systematically study 1) *The pedagogical demands of "Thought-Provoking" teaching* and empower *SocraticLM* to fulfill these demands. 2) *The teaching abilities of teachers* and reinforce these abilities in *SocraticLM*. 3) *The cognitive states of students* and enable *SocraticLM* to accurately identify them during the teaching process. Consequently, our *SocraticLM* can provide higher quality guidance that is more tailored and appropriate for each student's needs, transitioning from a "guardian of knowledge" to "choreographer of learning".

To build *SocraticLM*, we first construct a new dataset, *SocraTeach*, which consists of 35K high-quality, fine-grained Socratic-style multi-round teaching dialogues grounded in mathematical problems. In constructing the dataset, we propose a novel *"Dean-Teacher-Student"* pipeline, implementing three LLM agents to simulate the key roles in authentic teaching scenarios: *Dean, Teacher*, and *Student*. The *Dean* is a director that oversees and refines the *Teacher*'s instructions before they are presented to the *Student*, ensuring that the whole teaching process adheres to the Socratic style. The *Teacher* actively and gradually guides the *Student* to solve a problem by generating Socratic instructions, inspired by classic pedagogical theories [13, 45]. The *Student* responds to the *Teacher*'s instructions, where we establish a student cognitive state system that simulates six kinds of students in classroom to cover real and diverse teaching scenarios. Through multiple rounds of *"Teacher-Student"* interaction under the supervision of *Dean*, a comprehensive Socratic teaching dialogue is formed. One step further, to enhance the diversity and robustness of our dataset, we summarize four types of student responses from real teaching scenarios and perform data augmentation to generate extra 22K single-round teaching dialogues, specifically tailored to enhance four corresponding crucial teaching abilities.

---

[2]https://www.mathgpt.com/
[3]https://edugpt.com/

We fine-tune ChatGLM3-6b [12] on our *SocraTeach* dataset to obtain *SocraticLM*. During this process, we elaborate three training strategies to improve the pedagogical abilities while ensuring the problem-solving capacity of *SocraticLM* simultaneously. In addition, we contribute a novel evaluation system encompassing five pedagogical dimensions for assessing the teaching quality of LLMs, which to the best of our knowledge, is the first exploration in this field. Experimental results show that our dataset can enhance the pedagogical performances of LLMs and the teaching quality of our *SocraticLM* surpasses GPT4 by more than 12%.

The contributions of this paper are:

- We present *SocraticLM*, a language model that achieves Socratic "Thought-Provoking" teaching paradigm. Experimental results show that its Socratic teaching quality exceeds GPT4 by 12%, while maintaining the good problem-solving ability of the original ChatGLM3-6b.

- We construct a new dataset *SocraTeach* that contains massive, fine-grained Socratic teaching dialogues. To construct *SocraTeach*, we propose a novel *"Dean-Teacher-Student"* multi-agent pipeline, in which we design an innovative supervisory role *Dean*, a cognitive state system to direct the *Student*'s behavior, and an enhancement in four teaching abilities for the *Teacher*. This pipeline is general and can be transferred to the teaching in other subjects.

- We develop a five-dimensional comprehensive evaluation system to assess the teaching quality of LLMs, which to the best of our knowledge, is the first attempt in the field.

## 2   Related Work

**LLMs-enhanced intelligent education.** Large language models (LLMs) revolutionize three typical applications of intelligent education, namely automatic generation of educational resources, instant assessment of student learning outcomes, and personalized teaching assistance [26, 29, 42]. For educational resources, there is a tendency to use LLMs to generate textbooks, exercises, etc., based on teaching goals and needs, providing teachers with richer inspiration [4, 16]. For students' outcomes, LLMs can analyze students' homework and exams to provide assessments and feedbacks on their learning progress [9]. As for the most concerned personalized teaching in this paper, one line of research uses general LLMs like ChatGPT to provide students with multi-level assistance [44, 51, 58] in multiple disciplines, such as writing [18], programming [5], and medical education [25]. By analyzing students' learning data and behavioral patterns, these LLMs also have the potential to design unique learning paths to help students learn more effectively [20]. Another line of research is to collect a large amount of teaching instructions to fine-tune large models (e.g., EduChat [10]), giving them targeted teaching capabilities such as problem solving and emotional support.

**Personalized teaching dialogue dataset.** Constructing teaching dialogues is the basis for building LLM-based personalized teaching systems. In the literature, early attempts relied on crowd-sourcing (e.g., CIMA [49]) or rules (e.g., AutoTutor [17]) to create authentic dialogues. Subsequently, researchers adopted human-computer collaborative approaches. For instance, QuizBot [46] leveraged semantic similarity algorithms to analyze real students' responses and provided adaptive question by predefined teaching workflow. However, these methods required substantial manual effort or were constrained by predefined teaching procedures, resulting in limited scalability and difficulty in covering diverse real-world teaching scenarios. Recently, with LLMs demonstrating advantages in synthetic data generation [28], utilizing them to assist in teaching dialogue generation has attracted much attention. However, existing research [51] suggests that GPTs make for a bad teacher, thus current efforts mainly use LLMs to simulate students with different backgrounds [41], personalities [41] and error types [40], followed by human teachers providing explanations. Nevertheless, this process still requires human involvement, resulting in the latest dataset MATHDIAL [40] containing only 3K samples. Besides, these datasets also lack a systematic investigation into Socratic teaching.

## 3   The *SocraTeach* Dataset

Pedagogical theories point out that there are two basic demands for Socratic teaching [13, 45]: 1) it is fundamentally dialogic, relying on conversations between teachers and students to facilitate learning; 2) it uses probing questions to actively engage students, promoting independent thinking and encouraging them to find answers themselves. In building our *SocraTeach* dataset to meet these

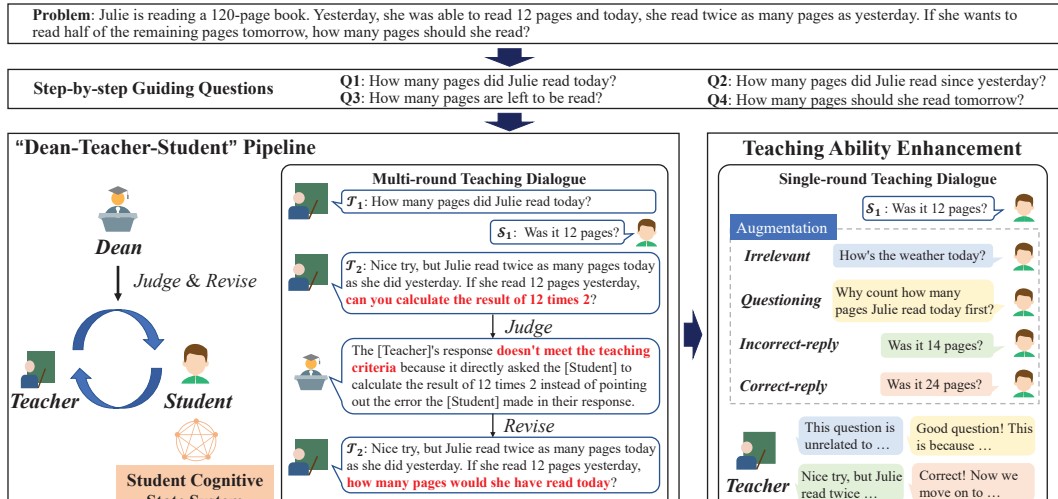

Figure 2: Workflow of our *SocraTeach* dataset construction.

requirements, we face the following challenges. First, for the teacher, there is considerable variability in pedagogical methodologies and presentation styles among teachers. It may be difficult for a model to learn all of them at once, which may result in confusion and errors within the teaching logic. Second, for the student, in real teaching scenarios, students' cognitive states are intricate and heterogeneous [22, 35]. While some students have strong understanding abilities and sufficient knowledge, there are also a considerable number of students who cannot understand the problem or even lack the essential knowledge. We expect that our dataset should cover all these situations, so that the model can learn to provide different levels of guidance for students with different states.

To solve the above challenges, we construct *SocraTeach* as follows. First, for each problem that needs to be taught, we decompose a list of step-wise guiding questions (Section 3.1). On this basis, we can control the simulation of teachers by aligning the instructional approach and explanatory style with these questions. Second, we devise an innovative *"Dean-Teacher-Student"* pipeline, implementing three LLM agents including *"Dean"*, *"Teacher"*, and *"Student"* to collect fine-grained multi-round teaching dialogues (Section 3.2). Especially, to align with student profiles in authentic scenarios, we build a cognitive state system to simulate six kinds of students in *"Student"* from the aspects of comprehension, calculation, knowledge mastery, etc. (Section 3.3). Finally, to further enhance the diversity and robustness of *SocraTeach*, we design data augmentation methods to construct additional single-round teaching dialogues for improving four crucial teaching abilities (Section 3.4).

## 3.1 Problem Collection & Step-by-Step Guiding Questions

In this paper, we take the teaching of mathematical problems at the primary school level as an example for exploration, because mathematics is a fundamental and critical subject and such problems involve the examination of students' basic understanding and reasoning abilities [32, 34]. Our problems are sourced from two representative datasets: MAWPS [27] and GSM8K [8], which contain 2.3K and 8.8K problems, respectively.

To ensure that the expression style and teaching approach are consistent in simulating the teacher role, we decompose each problem into a series of step-by-step guiding questions, such as **Q1**-**Q4** in Figure 2 (please refer to Appendix A for details). It should be noted that to ensure the efficiency and conciseness of teaching, a numerical calculation and a summary of the solution do not count as a step.

## 3.2 The *Dean-Teacher-Student* Pipeline

To create our *SocraTeach* dataset that achieves the Socratic "Thought-Provoking" teaching paradigm, we propose a novel "*Dean-Teacher-Student*" (DTS) pipeline to collect one-to-one, multi-round, teacher-student dialogues, which consists of three LLM agents:

- *Dean* $\mathcal{D}$: Research has indicated that GPTs have inadequacies in understanding students and language expression required to serve as a teacher [51]. To address this issue, we propose a

*Dean* agent to serve as an oversight role, which judges whether the *Teacher*'s instructions meet the requirements of Socratic teaching. If it thinks the instructions do not meet the requirements, it has the authority to revise them before they are presented to the *Student*.

- **Teacher $\mathcal{T}$**: The *Teacher* agent actively provokes the *Student* agent to solve problems in a Socratic style, serving two primary purposes according to Socrates' educational theory [13, 45]. First, it should prompt the *Student* to think at the appropriate time with Socratic questions, such as guiding the *Student* to consider the next step after completing a reasoning step. Second, it needs to provide the *Student* with explanations of the steps and the involved knowledge points. To maintain a consistent teaching style for a given problem, the *Teacher* is asked to deliver teaching following the step-by-step questions constructed in Section 3.1.

- **Student $\mathcal{S}$**: Representing the learner within the dataset, the *Student* agent generates replies to the *Teacher*'s instructions (i.e., questions and explanations). To ensure the authenticity and diversity of *Student*, we build a cognitive state system that describes six kinds of real students in Section 3.3 and set *Student* to simulate one of them each time it replies.

In DTS pipeline, each teaching dialogue $\{(\mathcal{T}_1, \mathcal{S}_1), (\mathcal{T}_2, \mathcal{S}_2), ...\}$ is formed by a cycle of interaction between *Teacher* and *Student* under the supervision of *Dean*, and each agent is simulated with GPT4. Taking Figure 2 as an example, in the first round ($t = 1$), the *Teacher* directly gives the question of the first step (i.e., $\mathcal{T}_1 = \mathbf{Q1}$) constructed in Section 3.1 (this process does not need the use of LLMs). Then, the *Student* selects a state profile (e.g., weak knowledge mastery) from six types of cognitive portraits in Section 3.3 and generates a corresponding response $\mathcal{S}_1$ based on it (see Appendix B.1 for prompt). After that, at $t = 2$, the *Teacher* provides instruction $\mathcal{T}_2$ in a Socratic style, in which we design pedagogical demands such as not providing answers but rather following the flow of the step-by-step questions. It should be emphasized that here we set up a different response style for each student profile through examples (Appendix B.2). After the *Teacher* generates $\mathcal{T}_2$, the *Dean* judges (e.g., "... doesn't meet the teaching criteria") and revises it (e.g., change "can you ... 12 times 2?" to "how many ... read today?"), focusing on 1) Whether it conforms to the Socratic style. 2) Whether it clearly points out the mistakes made by the *Student*. 3) Whether its language style resembles that of a real teacher (Appendix B.3), i.e., $\mathcal{T}_2 \leftarrow D(\mathcal{T}_2)$. The revised response is then sent to the *Student*, and the next round of dialogue begins. Ultimately, if the *Teacher* thinks that the teaching process has been completed, it will output an "[END]" token as the ending of its output, indicating to terminate the cycle. It is worth noting that although we focus on teaching mathematical problems in this paper, our DTS pipeline is general and can be extended to problems in other subjects (e.g., physics).

### 3.3 Student Cognitive State System

To ensure that our dataset covers the real and diverse student status throughout the teaching process, it is necessary to simulate different student cognitive states within the *Student* agent. However, a systematic and unified definition of these states has not been established in existing research [15]. Some previous studies have concentrated on states that are specific to particular subjects such as math and English [3, 11, 19, 50], while others abstractly define general states based on human cognitive science, such as concentration, working memory, and logical reasoning [14, 47, 53, 33]. Unfortunately, these definitions are either unadaptable to the teaching process or difficult to implement with LLMs. To address this issue, we review the Socratic teaching process from the perspective of students as follows. Initially, a student needs to grasp the meaning of the problem at hand. Then, he/she comprehends the instructions provided by the teachers and utilizes the computational ability and acquired knowledge to execute the instructions. Ultimately, this process fosters an interest in learning and helps to cultivate effective study results. Based on this idea, we summarize five dimensions of cognitive state:

(1) **Problem Understanding**: refers to the degree to which students understand the given problem.

(2) **Instruction Understanding**: refers to the degree to which students understand and carry out the teacher's instructions. A student in a good state should easily accomplish these instructions.

(3) **Calculation**: refers to the ability to derive mathematical expressions and numbers correctly.

(4) **Knowledge Mastery**: refers to the extent to which students have mastered knowledge.

(5) **Thirst for Learning**: refers to the students' desire or inclination to seek and acquire new information, ask questions, and explore possibilities.

Basically, we can define five types of students who perform poorly on one of the above dimensions. Moreover, we add a sixth type of student who excels at all dimensions.

## 3.4 Teaching Ability Enhancement

The multi-round dialogues constructed by DTS pipeline (denoted as $Dia_M$) ensure a model to grasp the fundamental Socratic teaching paradigm. However, in $Dia_M$, the *Student* responds only once to each instruction given by the *Teacher* and tends to choose simpler student portraits, leading to a lack of simulation for long-tail student responses (as discussed in Appendix D). In this section, to further enhance the diversity and robustness of our dataset, we construct more *Student-Teacher* single-round dialogues $Dia_S$ through data augmentation on $Dia_M$, improving four important teaching abilities.

Specifically, in real teaching processes, students' responses can be classified as follows. First, from the macro perspective, the responses can be divided into *"Irrelevant"* and *"Relevant"*. *"Relevant"* refers to responses directly related to the problem or instruction, while *"Irrelevant"* means that the responses are unrelated to the instructional content, such as asking "How's the weather today?" in Figure 2. Second, within the *"Relevant"* category, it can be further divided into *"Questioning"* and *"Replying"*, which refers to students asking questions to the teacher and answering the teacher's questions, respectively. Third, *"Replying"* can be further classified as *"Incorrect-reply"* and *"Correct-reply"* based on whether the students' responses to the teacher's question are correct or not. Along this line, students' responses include four categories: *"Irrelevant"*, *"Questioning"*, *"Incorrect-reply"*, and *"Correct-reply"*. On this basis, there are four key teaching abilities that need targeted enhancement.

First, for *"Irrelevant"* responses, we expect a teacher to recognize them and redirect the conversation towards teaching, such as responding "This question is unrelated to ... Let's focus on the problem first ...". To achieve this, we collect 200 genuine student inquiries from MOOCs that are unrelated to teaching and then construct $2,000$ single-round *Student-Teacher* dialogues by randomly inserting them into $Dia_M$ and asking *Teacher* to refuse answering (please refer to Appendix C for details).

Second, *"Questioning"* corresponds to the most crucial teaching ability, that is, a teacher should provide students with accurate explanations. Regarding it, we randomly sample $2,000$ *Teacher-Student* conversation $(\mathcal{T}_i, \mathcal{S}_i)$ from $Dia_M$ and use *Student* agent to ask three more questions $\mathcal{S}_i^1, \mathcal{S}_i^2, \mathcal{S}_i^3$ for $\mathcal{T}_i$ (see Appendix C.1 for prompt). Then, we ask the *Teacher* agent to provide $\mathcal{T}_{i+1}^1, \mathcal{T}_{i+1}^2, \mathcal{T}_{i+1}^3$, ultimately forming $6,000$ single-round *Student-Teacher* $\{(\mathcal{S}_i^j, \mathcal{T}_{i+1}^j)|j=1,2,3\}$ dialogues.

Third, a teacher should accurately identify students' *"Incorrect-reply"* and point out the idea of correction. To achieve this, we similarly sample $2,000$ *Teacher-Student* conversation from $Dia_M$ and employ rules and generation techniques to rewrite the *Student*'s responses into five wrong answers. Then, we explicitly prompt the *Teacher* to identify the errors and provide a response, obtaining another 10K instances of *Student-Teacher* dialogues (please refer to Appendix C for details).

Finally, to enable teachers to identify different expressions of the same *"Correct-reply"* for enhancing robustness, we take the same $2,000$ single-round "*Teacher-Student*" dialogues used for *"Incorrect-reply"* and create two correct responses with *Student* (see Appendix C.2 for prompt). Subsequently, we collect *Teacher*'s replies and obtain another $4,000$ single-round "*Student-Teacher*" dialogues.

## 3.5 Dataset Overview

In summary, our *SocraTeach* consists of 35K multi-round dialogues $Dia_M$ constructed by "*Dean-Teacher-Student*" pipeline in Sections 3.2 and 22K single-round dialogues $Dia_S$ through data augmentation in Section 3.4. The average number of rounds in $Dia_M$ is $5.28$, resulting in a total of 208K single-round dialogue examples. More statistics of *SocraTeach* are summarized in Appendix D.

In comparison to existing teaching dialogue datasets [17, 40, 41, 46, 49], our *SocraTeach* first addresses the deficiency of LLMs inadequately simulating teachers [51] by introducing the role of "*Dean*" for supervision and correction. Secondly, to the best of our knowledge, *SocraTeach* is the first publicly available dataset designed for Socratic teaching, which specifically enhances four key teaching abilities of *Teacher*. Thirdly, while existing datasets simulate different students by setting their demographic backgrounds (e.g., grade) or specific error types, *SocraTeach* models six cognitive states of *Student* during the teaching process based on pedagogical experience, which covers a wider range of authentic teaching scenarios and enables LLMs to possess better teaching capabilities. Lastly, *SocraTeach* is a fully automatically generated large-scale dataset containing 35K

multi-round dialogues and 22K single-round dialogues, significantly surpassing the existing datasets that rely on real human students/teachers (e.g., the latest MATHDIAL [40] contains 3K dialogues).

# 4 Fine-tune *SocraticLM*

Based on *SocraTeach*, we can fine-tune a SOTA LLM (e.g., ChatGLM3-6b [12]) by splitting each dialogue $\{(\mathcal{T}_1, \mathcal{S}_1), (\mathcal{T}_2, \mathcal{S}_2), ...\}$ into multiple rounds, using the preceding context of each round $\{(\mathcal{T}_1, \mathcal{S}_1), ..., (\mathcal{T}_i, \mathcal{S}_i)\}$ as input and the *Teacher*'s response $\mathcal{T}_{i+1}$ as output. However, it may lead to catastrophic forgetting and reduce the problem-solving ability that the model already has because these dialogues may differ from the data used for pre-training [21, 30, 38]. Specifically, we observe a decrease of 31.2%/9.7% in the accuracy of *SocraticLM* on the GSM8K/MAWPS dataset in Section 6.2. Therefore, to enhance *SocraticLM*'s teaching ability without compromising its fundamental problem-solving capability, we explore the following three training strategies:

**Separate Training.** To maintain problem-solving ability, one direct way is to mix the dialogue and problem-solving data for training. However, we discover that it does not yield satisfactory results as shown in Section 6.2. Therefore, we adopt a separate training approach wherein we first fine-tune *SocraticLM* using dialogue data and then fine-tune it on a small amount of problem-solving data randomly sampled from GSM8K and MAWPS. Our experiments revealed that optimal performance is achieved when the ratio $\alpha$ of problem-solving data to the dialogue data is approximately $\frac{1}{10}$.

**Instruction Tuning.** Inspired by [37], we employ different instructions for the dialogue data and problem-solving data, with the templates presented in Appendix E. It is worth noting that, unlike the prompt for *Teacher* in Section 3.2, here our instruction for dialogues does not require the model to follow the step-by-step guiding questions in Section 3.1. This is because providing such information in training may lead the model to take shortcuts, that is, to simplify the teaching process into information extraction from the prompt, without truly mastering the pedagogical ability.

**Mixed Prompt Setting.** Training with mixed prompt settings for the same task is an important method for improving LLMs' reasoning abilities [7, 55]. To this end, in addition to the original zero-shot problem-solving data of GSM8K and MAWPS, we also construct their one-shot version for training, which consists of approximately $\frac{1}{10}$ of the amount of zero-shot data.

# 5 Our Socratic Teaching Evaluation System

Since there is no standard answer for the teaching process, previous metrics that calculate the similarity between model-generated responses and annotated responses (e.g., BLEU [43], Rouge [31]) may not fully assess the teaching quality of LLMs. To address this issue, in this paper, we contribute an evaluation system encompassing five pedagogical dimensions for Socratic style and teaching abilities, which to the best of our knowledge, is the first comprehensive exploration in this field.

(1) *Overall Quality (Overall)*: This metric is a holistic and subjective evaluation of teaching quality, requiring that the instruction satisfies Socratic style and enhances students' experience.

For *Overall Quality*, we randomly select $1,000$ single-round "*Student-Teacher*" conversations from $Dia_M$ and recruit 10 well-educated annotators to blindly rank the *Teacher* response pairs provided by each model and GPT4 in the same context (please refer to Appendix F for details). The *Overall Quality* is estimated by a normalized win rate difference $\frac{1}{2}(1 + \frac{Win-Lost}{Win+Lost+Tie}) \in (0,1)$ (GPT4's own result is $0.5$). To ensure agreement of quality judge among humans, we also randomly construct 100 *Teacher* response pairs of *SocraticLM* and GPT4 and ask all annotators to judge which one is better. The Kappa score is $0.70$, indicating good agreement among human annotators.

For the four Socratic teaching abilities we elaborated in Section 3.4, we propose metrics (2)-(5). For each of them, we randomly select 100 corresponding single-round dialogues from $Dia_S$ for testing.

(2) *Incorrect Answer Recognition Accuracy (IARA)*: This dimension focuses on whether the teacher can accurately identify students' *"Incorrect-reply"*. For example, in Figure 2, if a student provides an incorrect answer (e.g., "14"), a competent teacher should be able to recognize and point it out. This process is objective and can be considered as a binary classification task.

| | Overall | IARA | CARA | SER | SRR | BLEU-4 | Rouge-1 | Rouge-2 | Rouge-l |
|---|---|---|---|---|---|---|---|---|---|
| ChatGPT | 0.29 | 0.42 | 0.93 | 0.62 | 0.19 | 22.8 | 34.3 | 14.4 | 21.3 |
| GPT4 | 0.50 | 0.76 | 0.91 | 0.65 | 0.55 | 36.2 | 42.9 | 19.4 | 32.4 |
| Vicuna-7b | 0.15 | 0.16 | 0.77 | 0.16 | 0.39 | 22.8 | 34.9 | 14.4 | 22.8 |
| Llama2-7b | 0.27 | 0.15 | 0.86 | 0.32 | 0.13 | 28.3 | 35.7 | 14.0 | 24.1 |
| Llama2-13b | 0.25 | 0.23 | 0.87 | 0.30 | 0.08 | 27.9 | 36.4 | 14.3 | 23.6 |
| Llama3-8b | 0.33 | 0.75 | 0.77 | 0.39 | 0.52 | 27.4 | 33.0 | 10.9 | 22.0 |
| ChatGLM3-6b | 0.11 | 0.18 | 0.87 | 0.46 | 0.07 | 17.1 | 26.2 | 9.1 | 15.7 |
| EduChat-32b | 0.37 | 0.48 | 0.77 | 0.40 | 0.03 | 27.8 | 38.4 | 17.9 | 29.2 |
| *SocraticLM* (**ours**) | **0.62** | **0.83** | **0.98** | **0.74** | **0.78** | **48.6** | **56.2** | **33.7** | **47.5** |
| w/o $Dia_S$ | 0.54 | 0.27 | 0.89 | 0.67 | 0.34 | 42.6 | 52.2 | 32.3 | 44.4 |
| w/o *Irrelevant* | 0.57 | 0.79 | 0.87 | 0.69 | 0.43 | 45.7 | 52.3 | 29.8 | 44.2 |
| w/o *Questioning* | 0.58 | 0.74 | 0.92 | 0.53 | 0.83 | 47.8 | 55.0 | 31.9 | 45.9 |
| w/o *Incorrect* | 0.51 | 0.33 | 0.93 | 0.68 | 0.65 | 41.8 | 48.2 | 30.4 | 38.9 |
| w/o *Correct* | 0.60 | 0.70 | 0.58 | 0.70 | 0.76 | 47.4 | 55.1 | 32.8 | 46.6 |

Table 1: Teaching performances. For all metrics, the higher value denotes the better performance. The best methods are highlighted in bold. The runner-up baselines are represented by underline.

(3) ***Correct Answer Recognition Accuracy (CARA)***: In contrast to error recognition, this dimension focuses on whether the model can accurately identify students' *"Correct-reply"*. Neglecting this metric may mislead the LLMs to consider any answer provided by the student as incorrect.

(4) ***Successful Explanation Rate (SER)***: This dimension focuses on whether the model can provide students with satisfactory explanations for their *"Questioning"*. This metric is subjective, but can be converted to binary classification based on students' real experience.

(5) ***Successful Rejection Rate (SRR)***: This metric is designed for the case where a teacher should refuse to answer students' *"Irrelevant"* questions and redirect them back to the instructional content. Based on whether the model refuses to answer the question, it is also calculated as binary classification.

Compared with existing works in evaluating LLMs for education, our evaluation system offers three main advantages. First, it provides a more comprehensive and adequate assessment. While previous works either rely on similarity metrics (e.g., BLEU [43]) or limited-scale manual evaluations (e.g., issuing questionnaires [18]), our system assesses overall teaching quality along with four key teaching abilities, which provides a more systematic organization of evaluation. Second, it enables better comparability across LLMs. Limited by the fact that a student can only interact with one LLM at a time, traditional human evaluations [5, 24] are difficult to compare the effectiveness of multiple models. In contrast, our system uses the same teaching dialogues shared across different models as test samples, allowing for a fair comparison of different LLMs simultaneously. Third, our system is more extensive and reliable benefiting from our larger *SocraTeach* dataset, while recent studies rely on smaller datasets, such as the latest one [40] with only around 600 testing dialogues.

## 6 Experiments

In this section, we verify the effectiveness of our *SocraticLM* by taking ChatGPT, GPT4, Vicuna-7b [6], Llama2-7b, Llama2-13b, Llama3-8b [52], ChatGLM3-6b [12], and EduChat-32b [10] as baselines. The implementation details are described in Appendix G. Especially, for fair comparison, for the problems taught in the testing dialogues, we omit all of their dialogues in training. In addition to our proposed evaluation system in Section 5, we also invite human annotators to give a real teacher response for each testing dialogue, taking it as a standard to calculate the BLEU and Rouge.

### 6.1 Main Results

Table 1 summarizes the results for all models. First, our *SocraticLM*, which contains 6 billion parameters, demonstrates a significant improvement in all Socratic teaching abilities. Notably, it outperforms GPT4 by 12% on *Overall*, 6% on *IARA*, 7% on *CARA*, 9% on *SER*, and 23% on *SRR*, as well as over 12% in mirroring the responses of human teachers as measured by BLEU and Rouge. In Appendix H, we present and analyze examples of their outputs. Second, our *SocraticLM* demonstrates a significant improvement in *SER*. This indicates that the judgement and correction of our proposed *Dean* agent can critically boost the explanatory capabilities of large models when they function

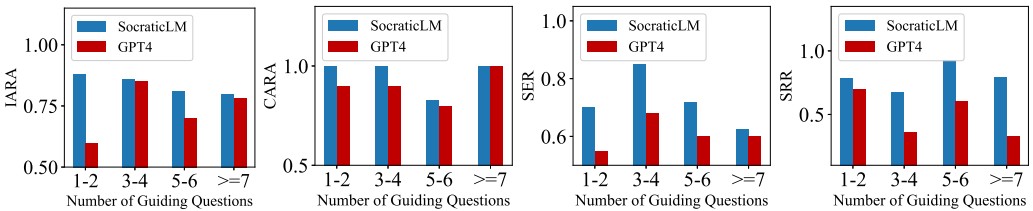

Figure 3: Performances on problems with different number of step-by-step guiding questions.

as teachers. Third, in Figure 3, we evaluate the four teaching abilities on problems with different numbers of step-by-step guiding questions, which can reflect the difficulty of the problems. It is evident that our *SocraticLM* consistently outperforms GPT4 across all difficulty levels.

## 6.2 Ablation Study

**Importance of Teaching Ability Enhancement.** We explore the importance of single-round teaching dialogues $Dia_S$ built for the four key teaching abilities in Section 3.4. From Table 1, we first observe a significant decline (e.g., *Overall Quality* by 8%) when these dialogues are removed (i.e., "w/o $Dia_S$"). This illustrates the necessity of our teaching ability enhancement and confirms that our proposed four teaching abilities are effective in meeting the real demands of Socratic teaching. Second, the *IARA* and *SRR* metrics decrease the most, indicating the greatest disparity between the current LLM-based teaching and human teaching may lie in the response to students' incorrect answers and irrelevant questions. Third, each time a type of single-round dialogue data is eliminated, all teaching abilities will show a decline, which indicates that there is a coupling effect among different teaching abilities. Especially, the *CARA* metric in the absence of dialogues for students' *"Correct-reply"* (i.e., "w/o *Correct*") is even lower than when all single-round data is removed ("w/o $Dia_S$"). We hold the reason may be that in this case, *SocraticLM* was still fine-tuned on dialogues corresponding to students' *"Incorrect-reply"*. This causes the model to develop a stronger tendency to perceive a student's reply as incorrect. This phenomenon further indicates that it is necessary to balance different types of single-round dialogues to avoid overfitting on specific instructional patterns.

**Importance of Ability-balancing Strategies.** Here we discard the problem-solving data and the three ability-balancing strategies in Section 4 in training to investigate their influence. From Table 2, fine-tuning without problem-solving data ("w/o *Problem*") will result in a 31.2%/9.7% lower accuracy on GSM8K/MAWPS compared with ChatGLM3-6b. This could be attributed to the notable differences between teaching dialogues and data used for LLM pre-training, causing a dramatic disturbance in the parameters. Besides, all three training strategies are effective. Among them,

|  | *Overall* | $ACC_G$ | $ACC_M$ |
|---|---|---|---|
| ChatGLM3-6b | 0.11 | 0.624 | 0.798 |
| *SocraticLM* | 0.62 | 0.606 | 0.814 |
| w/o *Problem* | 0.58 | 0.312 | 0.701 |
| w/o *Separate* | 0.54 | 0.159 | 0.646 |
| w/o *Instruction* | 0.02 | 0.320 | 0.625 |
| w/o *Mixed-Prompt* | 0.56 | 0.605 | 0.804 |

Table 2: Performance without problem-solving data and three ability-balancing training strategies in Section 4. $ACC_G$, $ACC_M$ represent the accuracy on GSM8K and MAWPS, respectively.

*Separate Training/Instruction Tuning* has the greatest influence on problem-solving/Socratic teaching respectively. *Mixed Prompt Setting* might have already been employed in the LLM pre-training, hence exhibiting less noticeable improvements. Moreover, it is worth noting that *SocraticLM* achieves higher accuracy on MAWPS than ChatGLM3-6b. We speculate the reason is that, through fine-tuning on our *SocraTeach* dataset, *SocraticLM* indeed learns to address multiple student questions about various aspects of a single problem (e.g., asking about each reasoning step and the knowledge involved). This process allows *SocraticLM* to develop a deeper understanding of the problem-solving process, which in turn can improve its problem-solving accuracy.

## 6.3 Influence of Data Scale

Data scale is crucial for both the efficiency and effectiveness of training large language models. In order to investigate this issue, in this section, we vary the amount of multi-round dialogues in *SocraTeach* dataset and the ratio $\alpha$ between multi-round dialogues and problem-solving data.

**Scale of Multi-round Dialogues.** To study the impact of different data scale, we randomly select 25%, 50%, 75% multi-round dialogues from *SocraTeach* and expand it to 125% dialogues by running DTS pipeline more times to train *SocraticLM*. The results in Figure 4 indicate that (i) Our data is not only effective but also impactful at different scales, which can significantly enhance the teaching capability of LLMs. (ii) As the volume of data increases, we observe a corresponding increase in the teaching ability. This correlation highlights the importance of data quantity in model performance. Specifically, it is noteworthy that a minimum of 75% ($\approx$ 26K) dialogues is required for surpassing the *Overall Quality* of GPT4. (iii) As the volume of data surpasses

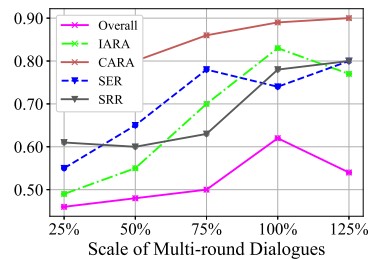

Figure 4: Effects of dialogue scale.

the 35K threshold, it tends to approach a saturation point where further increases in data volume yield smaller incremental benefits to the model's capability. Specifically, at the 125% data scale, the *IARA* metric shows a decline, indicating that the root cause of this saturation is a decrease in the model's ability to identify incorrect answers (the decline in *Overall Quality* is a subsequent result). This may be because, with the increase in multi-round dialogue data, the proportion of single-round dialogue data for *"Incorrect reply"* decreases. When multi-round data scale exceeds 125%, this proportion may fall below a certain threshold, which results in diminishing effectiveness.

**Scale of Problem-solving Data.** Figure 5 shows the performance changes of *SocraticLM* as we adjust the ratio $\alpha$ between problem-solving data and dialogue data. The trend indicates that having too few or too much problem-solving data does not lead to satisfactory problem-solving ability. Instead, a balance needs to be struck with the teaching dialogue data. In fact, an excessive introduction of problem-solving data may even result in 1.9% decrease in accuracy on GSM8K. This could be attributed to that the parameters corresponding to *SocraticLM'* problem-solving ability might undergo disturbances after initial fine-tuning using the teaching dialogues. When retraining with problem-solving data, it requires to re-strike a delicate balance between underfitting and overfitting of this ability.

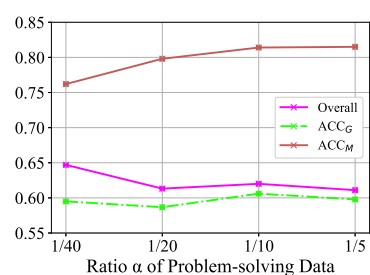

Figure 5: Effects of problem-solving data scale.

# 7 Conclusion

In this paper, we introduced *SocraticLM*, a LLM designed to facilitate Socratic "Thought-Provoking" personalized teaching. To build *SocraticLM*, we proposed a "*Dean-Teacher-Student*" pipeline to construct *SocraTeach* dataset, which simulated six student cognitive states and strengthened four crucial teaching abilities. Besides, we developed a comprehensive teaching ability evaluation system for LLMs. Experiments demonstrated that *SocraticLM* significantly outperforms current LLMs such as GPT4 and validated the necessity of each component within the *SocraTeach* dataset. We discuss more cases, the broader impacts, limitations, and future work in Appendix H, I, and J.

## Acknowledgments and Disclosure of Funding

This research was partially supported by grants from the National Key Research and Development Program of China (No.2021YFF0901005), National Natural Science Foundation of China (No.62477044, 62337001), and the Key Technologies R&D Program of Anhui Province (No.202423k09020039).

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

# A More details about problem decomposition

A decomposition of each problem was provided in the GSM8K raw data, which we found largely met our needs. Thus, we use GPT4 to decompose the problems in MAWPS with the prompt in A.1.

---

**A.1: Problem Decomposition Prompt**

You are a math teacher, and I want you to answer the math problems I give you in steps. In each step, you need to first give a guided question and then explain the process of solving it, meeting the following criteria:
- After all the steps, the final result is the same as the Answer
- The steps need to be as few as possible
- Not a calculation process as a step
- Not a result summary as a step
- Different steps are divided by '\n'
- Please ensure that your steps are well-organized, precise, and easy to follow, with each step building upon the previous one.

#Problem: Natalia sold clips to 48 of her friends in April, and then she sold half as many clips in May. How many clips did Natalia sell altogether in April and May?
#Answer: 72
#Steps: How many clips did Natalia sell in May? ** Natalia sold 48/2 = 24 clips in May.\nHow many clips did Natalia sell altogether in April and May? ** Natalia sold 48+24 = 72 clips altogether in April and May.

#Problem: {Here is the target problem.}
#Answer: {Here is the target answer.}

---

# B Prompts of "*Dean-Teacher-Student*" pipeline

---

**B.1: Prompt of *Student* Agent**

You are a primary school student, please refer to the dialogue and generate a possible response for me. In generating a response, you first need to choose one of the following SITUATIONs and then respond as a student who fits that situation, meeting the following criteria:
- You are not a teacher, do not give any teacher reply!
- Don't ask questions about the people in the question.
- Your response should cohere with the given Dialogue.
SITUATIONs: (1) a student does not understand the meaning of the problem; (2) a student does not understand the content explained by the teacher; (3) a student makes error in calculation; (4) a student has poor knowledge mastery; (5) a student has weak thirst for knowledge; (6) a student has strong abilities in all aspects.

**Here are some examples:**
#Problem: Natalia sold clips to 48 of her friends in April, and then she sold half as many clips in May. How many clips did Natalia sell altogether in April and May?
#Dialogue: [Teacher]How many clips did Natalia sell in May?\n[Student]I'm not good at math. Can you explain it again?\n[Teacher]Not a problem at all. Let's break it down. If Natalia sold half as many clips in May as she did in April, and she sold 48 clips in April, then how many clips will she have sold in May? Let's take the number from April and half it. What do you get?
#Response:Choose SITUATION (3)[Student]I get it, the half of 48 is 240.
…

**Here is the target problem:**
#Problem: {Here is the target problem.}
#Dialogue: {Here is the target dialogue.}

---

## B.2: Prompt of *Teacher* Agent

I am a primary school student. You are a teacher that always responds in the Socratic style. You could not give me the answer but always try to ask just the right question to help me learn to think for myself. You need to break the problem down into simpler parts until it's at the right level for me. Please explain the above steps to me in the form of questions, starting with [Step 1]. If you think you have finished teaching, please put \"[END]\" at the end of your response.

**(Choose one of the following examples according to *Student*'s SITUATION):**
**(SITUATION 1)**
#Problem: Natalia sold clips to 48 of her friends in April, and then she sold half as many clips in May. How many clips did Natalia sell altogether in April and May?
#Steps: [step 1] How many clips did Natalia sell in May?\n[step 2] How many clips did Natalia sell altogether in April and May?
#Dialogue:[Teacher]: How many clips did Natalia sell in May?\n[Student]: I'm confused. Does \"half as many\" mean the same number as April?
#Response:[Teacher]: In fact, \"half as many\" means half of the number. So if Natalia sold 48 clips in April, how many clips did she sell in May?

**(SITUATION 2)**
#Problem: Randy has 60 mango trees on his farm. He also has 5 less than half as many coconut trees as mango trees. How many trees does Randy have in all on his farm?
#Steps: [step 1] How many mango trees does Randy have?\n[step 2] How many coconut trees does Randy have?\n[Step 3] How many trees does Randy have in all on his farm?
#Dialogue:[Teacher]: How many mango trees does Randy have?\n[Student]: You mean how many are there totally or just mango?
#Response:[Teacher]: We're talking about just the mango trees now. So, considering just the mango trees, can you tell me how many Randy has according to the problem?

**(SITUATION 3-6)**
…

**Here is the target problem:**
#Problem: {Here is the target problem.}
#Steps: {Here is the target steps.}
#Dialogue: {Here is the target dialogue.}

## B.3: Prompt of *Dean* Agent

You are a teaching director. The following is the teaching process of a Socratic teacher. Please evaluate the response of [Teacher] in the last round and judge whether it meets the following teaching criteria:
- If the [Student] is asking a commonsense or struggles to understand a concept, the [Teacher] is allowed to give direct explanation.
- In other cases, no direct answer or solution to the problem can be given in a declarative or rhetorical manner.
- If [Student] makes a mistake, it is necessary to imply by way of question what error he/she made.
- [Teacher] needs to speak like a teacher and does not use phrases like \"let me ask you\".
- The response must be smooth and fluent.

If you think the response meets the criteria, please reply [True], otherwise reply [False], explain your judgement and modify it according to the criteria.

**Here are some examples:**
#Problem: Natalia sold clips to 48 of her friends in April, and then she sold half as many clips in May. How many clips did Natalia sell altogether in April and May?
#Dialogue:[Teacher]: How many clips did Natalia sell in May?\n[Student]: I'm confused. Does \"half as many\" mean the same number as April?\n[Teacher]: Not exactly, \"half as many\" means half the amount. If Natalia sold 48 clips in April, what would be half of 48? That would tell us how many she sold in May. Can you solve that?
#Response:[**True**].

#Problem: Natalia sold clips to 48 of her friends in April, and then she sold half as many clips in May. How many clips did Natalia sell altogether in April and May?
#Dialogue:[Teacher]: How many clips did Natalia sell in May?\n[Student]: Is it 48 plus 48 divided by 2? So, 72?\n[Teacher]: You're on the right track, dividing by 2 to figure out the number of clips sold in May. But let me ask you this: does the 48 add to 48 divided by 2, or do you divide 48 by 2 on its own first?
#Response:[**False**].The [Teacher]'s response is incorrect and does not point out that the [Student] made a calculation error. [**Modified Teacher**]: You're on the right track! But you may make a mistake in calculations. Can you calculate again what is the result of 48 plus 48 divided by 2?

#Problem: Betty is 60 years old, and she is the oldest person in the family. Her daughter is 40 percent younger than she is, and her granddaughter is one-third her mother's age. How old is the granddaughter?
#Dialogue:[Teacher]: How old is Betty's daughter?\n[Student]: 60 percent of 60 is... 36, so 36?\n[Teacher]: You're on the right track thinking about percentages, but remember when we say someone is \"40 percent younger\", we need to subtract that percentage from 100%. Can you revisit your calculation with this in mind?
#Response:[**False**]. The [Student] correctly gets the answer of "How old is Betty's daughter" but the [Teacher] still asks the same question. [**Modified Teacher**]: Yes! You have got the correct result of Betty's daughter's age. Now you can calculate how old is Betty's granddaughter. Can you have a try?

**Here is the target problem:**
#Problem: {Here is the target problem.}
#Dialogue: {Here is the target dialogue.}

## C    More Details about Teaching Ability Enhancement

To build single-round dialogues for "*Irrelevant*" student response, we first randomly select $2,000$ dialogues from $Dia_M$ constructed by DTS pipeline. From each dialogue, we randomly select one round of *Student*'s response and replace it with a randomly selected question from the 200 questions collected from MOOCs. We then use *Teacher* agent to refuse answering the question under the supervision of *Dean*, finally forming $2,000$ single-round *Student-Teacher* dialogues.

---

**C.1: Prompt of *Questioning***

#Problem: {Here is the target problem.}
#Answer: {Here is the target answer.}
#Analysis: {Here is the target analysis.}
#Dialogue history: {Here is the target dialogue history.}

You are a primary school student, please for the last round of [Dialogue history], put forward three questions that primary school students may ask in concise language. Different questions should be separated by ** .

---

To build five wrong answers for "*Incorrect-rely*", on one hand, we identify all the numbers (e.g., "2") and operators (e.g., "+") in a *Student*'s reply and randomly introduce a perturbation within the

| | |
|---|---|
| Number of Problems | 11,147 |
| Number of Multi-round Dialogues in $Dia_M$ | 35,151 |
| Number of Single-round Dialogues in $Dia_S$ | 22,000 |
| Total Number of Single-round Dialogues | 207,581 |
| Maximum / Minimum Number of Rounds | 12 / 3 |
| Average Number of Step-by-step Guiding Questions | 3.29 |
| Average Number of Rounds in $Dia_M$ | 5.28 |
| Average Length of *Student*'s response | 16.4 |
| Average Length of *Teacher*'s response | 30.3 |

Table 3: Statistics of our *SocraTeach* dataset.

range of 10 to one number or randomly replace one operator with another. We apply these rules to obtain two new *Student*'s responses. On the other hand, we use GPT4 to rewrite the original *Student*'s response and generate three incorrect answers using prompt in C.2. With these five responses, we prompt the *Teacher* to response and obtain 10K *Student-Teacher* dialogues.

# D Statistics of *SocraTeach* Dataset

As stated in Section 3.5, our *SocraTeach* consists of 35K multi-round dialogues $Dia_M$ and 22K single-round dialogues $Dia_S$. The average number of rounds in $Dia_M$ is 5.28. On average, the *Student/Teacher*'s responses contain 16.4/30.3 words respectively. The overall statistics is summarized in Table 3. Moreover, from Figure 6(a), most teaching dialogues consist of 5-6 rounds. In Figure 6(b), we visualize the probability mass function of *Student*'s cognitive states in *SocraTeach*, where (1)-(5) correspond to a student portrait that performs poor in one of the dimensions in our proposed student cognitive system (Section 3.3), while (6) corresponds to a student with strong states in all dimensions. It can be seen that the *Student* agent tends to simulate a student who is excellent in all aspects or alternatively, has problems in calculation ability (i.e., "(3)") or knowledge mastery (i.e., "(4)").

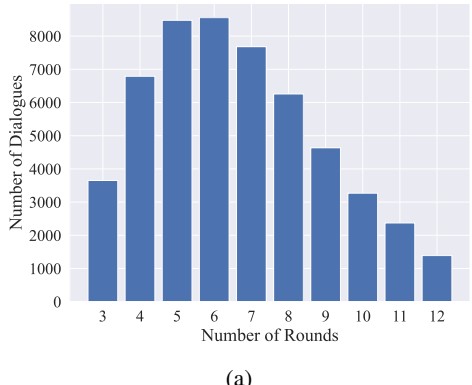

(a)

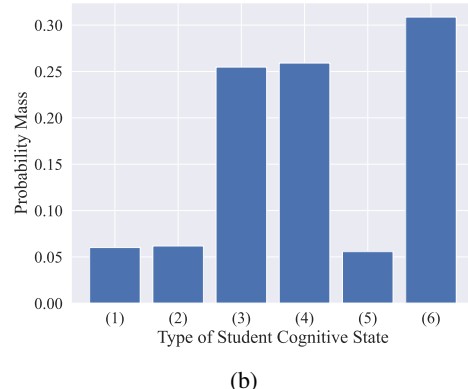

(b)

Figure 6: Distributions of Number of Rounds (a) and Student Cognitive State (b).

# E  Instruction Tuning Template

Figure 7 shows the instruction template for teaching dialogue data and problem-solving data.

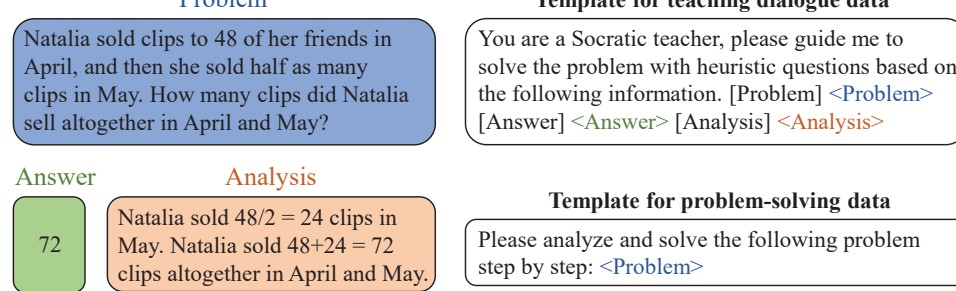

Figure 7: The templates for instruction tuning.

# F  Human Evaluation for Teaching Quality

To evaluate the *Overall Quality* of different LLMs, we recruit 10 well-educated annotators to rank the instructions provided by each model compared with GPT4. Figure 8 shows the template.

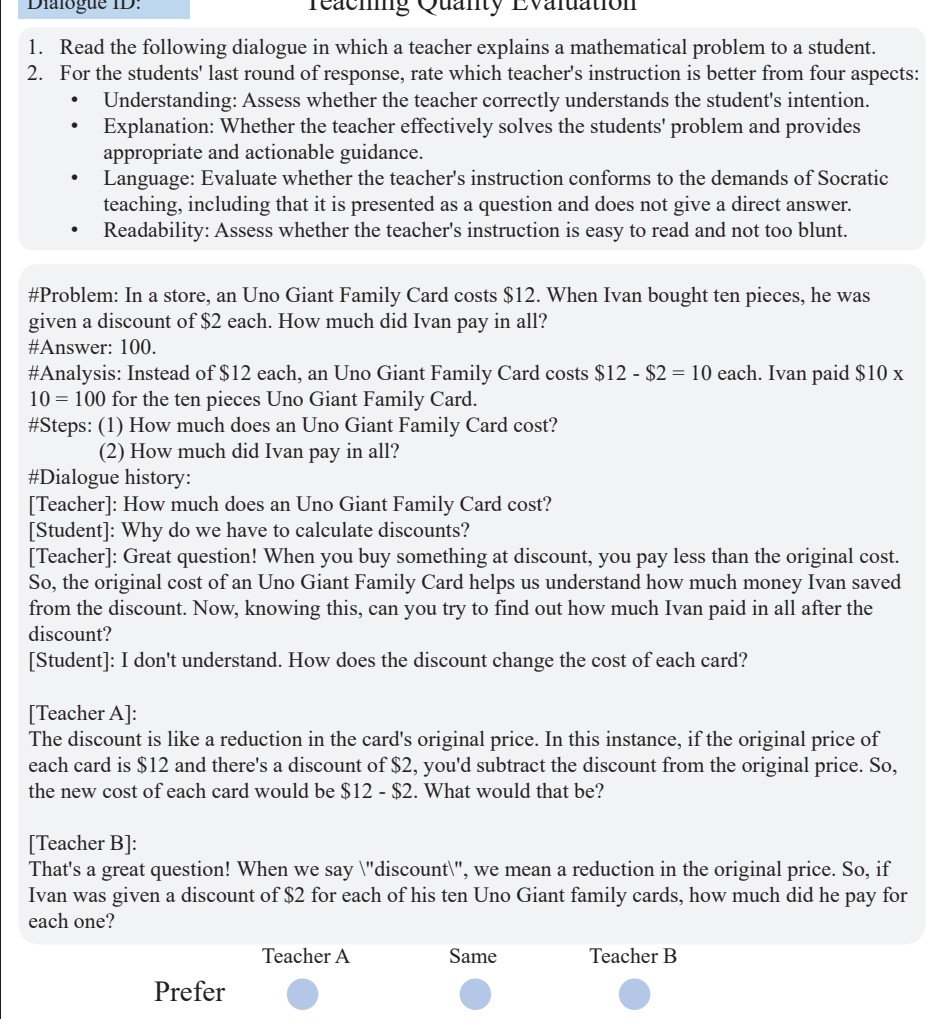

Figure 8: The template of human evaluation for *Overall Quality*.

# G   Implementation Details

Our *SocraticLM* is obtained by P-Tuning [36] ChatGLM3-6b (not ChatGLM3-6b-Base) for 2 epochs with a learning rate of 0.02 and batch size of 64. For dataset partition, as described in Section 5, the testing set is composed of 400 single-round dialogues for evaluating objective teaching ability (i.e., *IARA, CARA, SER, SRR*) and 1,000 multiple-round dialogues for evaluating subjective *Overall Quality*. Of the remaining data in *SocraTeach*, 10%/90% is used for validation/training. All experiments are conducted on a server with six NVIDIA RTX 3090 GPUs. Our dataset and code is available at https://github.com/Ljyustc/SocraticLM.

# H   Additional Model Samples

In the following, we present four samples of our *SocraticLM* and GPT4 for students' "*Incorrect-reply*", "*Correct-reply*", "*Questioning*" and "*Irrelevant*" response respectively. In the first example, both *SocraticLM* and GPT4 identify the student's mistake, but GPT4 gives the answer "7" directly, which contradicts the demands of Socratic teaching. In the second example, GPT4 also gives an almost complete solution process, which does not meet the requirements of step-by-step Socratic guidance of thinking. In the third example, GPT4's expression is more stilted and lacks the sense of a real human teacher. In the fourth example, *SocraticLM* redirects the student to solve the math problem first, while GPT4 is shifted to answer the irrelevant question.

---

**Sample 1: Incorrect-reply**

#Problem: Brenda's mother made cookies for 5 . If she prepared 35 cookies and each of them had the same number of cookies , how many did each of them have ?
#Dialogue history:
[Teacher]: How can we calculate the number of cookies each person gets?
[Student]: Can't you just divide it for me?
[Teacher]: Well, you could just do that, but it's important to understand why we divide. Dividing allows us to equally distribute something amongst a certain number of people or things. In this case, if Brenda's mother made 35 cookies for 5 people, how would you divide those cookies so that each person gets the same amount?
[Student]: Oh, do we just divide the total number of cookies by the total number of people?
[Teacher]: That's right! Now using this method, how many cookies does each person get if Brenda's mother made 35 cookies for 5 people?
[Student]: I'm not sure. What is 35 divided by 5?
[Teacher]: Good question! Let's think about it. If you have 5 groups, and you need to distribute 35 items equally, how many would go in each group? You can start by seeing how many times 5 can fit into 35 without going over. Can you try to do that?
[Student]: So does that mean each person gets 8 cookies?

- - - - - - - - - - - - - - - - - - - - - - - - - - - - - - - - - - - - - - - - - - - - - - - - -

**SocraticLM:**
That's a good start! But let's double-check. If you divide 35 by 5, what do you get?

**GPT4:**
Not quite. If you visualize a group of 35 objects, and you divide it into 5 equal groups, you would see that each group actually contains 7 objects. In this case, each person would get 7 cookies. Does that make sense?

---

**Sample 2: Correct-reply**

#Problem: Jeff makes 10 donuts each day for 12 days. Jeff eats one of the donuts each day. Chris then comes over and eats 8 donuts. If 10 donuts fit in each box, how many boxes can Jeff fill with his donuts?
#Dialogue history:
[Teacher]: How many donuts does Jeff make?

[Student]: Does \"makes\" include the ones he eats or not?

[Teacher]: When we say \"Jeff makes\", we're referring to the total number of donuts Jeff creates, whether he eats them or not. So, how many donuts in total did Jeff make?

[Student]: Jeff makes 10 donuts each day and he made donuts for 12 days, so he made 120 donuts in total.

[Teacher]: That's correct! Now, how many donuts does Jeff eat?

[Student]: Jeff eats 12 donuts because he has one each day and he made donuts for 12 days.

- - - - - - - - - - - - - - - - - - - - - - - - - - - - - - - - - - - - - - - - - - - - - - - - - -

**SocraticLM:**

Excellent! Now, if Jeff eats 12 donuts and he made 120 donuts in total, how many donuts are left after he eats?

**GPT4:**

Exactly! Now, there is another character introduced in the problem named Chris. How many donuts does Chris eat? Given that Chris eats 8 donuts, how we calculate the total number of remaining donuts? Once we have the total number of remaining donuts, what do we do to find out the number of boxes required? Now, can you calculate the number of boxes that can Jeff needs to store all of his remaining donuts?

---

## Sample 3: Questioning

#Problem: Matias is a salesman in a bookstore. He sold 7 books on Tuesday and three times as many on Wednesday. If the sales from Wednesday were tripled on Thursday, how many books did Matias sell during these three days combined?

#Dialogue history:

[Teacher]: How many books did Matias sell on Wednesday?

[Student]: What are you asking?

- - - - - - - - - - - - - - - - - - - - - - - - - - - - - - - - - - - - - - - - - - - - - - - - - -

**SocraticLM:**

Let's break it down. If Matias sold 7 books on Tuesday, and he sold three times as many on Wednesday, how many books did he sell on Wednesday?

**GPT4:**

How many books did Matias sell on Tuesday? And if he sold three times more on Wednesday, can you calculate how many books that would be?

---

## Sample 4: Irrelevant

#Problem: Solomon collected three times as many cans as Juwan. Levi collected half of what Juwan collected. Solomon collected 66 cans. How many cans did the boys collect in all?

#Dialogue history:

[Teacher]: How many cans did Juwan collect?

[Student]: How to study grammar?

**SocraticLM:**

That's a great question! But let's first figure out how many cans Juwan collected. If Solomon collected three times as many cans as Juwan, and Solomon collected 66 cans, can you calculate how many cans Juwan collected?

- - - - - - - - - - - - - - - - - - - - - - - - - - - - - - - - - - - - - - - - - - - - - - - - - -

**GPT4:**

That's an entirely different topic, but I can help. Study of grammar means understanding the structure of a language. We have verbs, nouns, adjectives, adverbs, etc. in a sentence. Understanding how these work together is studying grammar. Have you tried reading about grammar before?

# I   Broader Impacts

In this paper, we propose a large language model *SocraticLM* to achieve Socratic teaching, which is a crucial pedagogy in intelligent education and has a significant impact on our daily lives. On the one hand, as explained in Section 3.3 and 3.4, our *SocraticLM* can provide instructions to students with different cognitive states and various types of responses. Therefore, it can support multiple personalized applications in classrooms and online platforms, delivering high-quality instructions more efficiently and conveniently to students from various backgrounds, thus promoting educational equity and rapid development. On the other hand, in our work, we use GPT4 to simulate conversations between teachers and students to generate dialogue data, entirely without the involvement of real human teachers or students. This approach provides a way to collect data without interacting with or testing on real humans, reducing the burden of human interaction while also avoiding privacy, security, and other ethical concerns. However, since our model needs to be trained with multi-round teaching dialogue data, in order to expand to more subjects (e.g., physics), it may require more educational resources (e.g., problems in textbooks) and training costs.

# J   Limitations and Future Work

First, from Table 1, the *SER* metric of *SocraticLM* and GPT4 is $0.74$ and $0.65$, respectively. This shows that the current models have room for improvement in their ability to respond to real and complex student questions. Second, we focus on the teaching of mathematical problems in this paper. For other subjects, we need additional data construction and training processes. Third, the testset of problem-solving ability in this paper is the problems that our Socratic teaching dialogues are based on (i.e. GSM8K and MAWPS). In order to more accurately assess the changes in reasoning ability, we will test additional datasets and explore ability-balancing training strategies for more tasks. Finally, in this paper, we use $\mathrm{ChatGLM3\text{-}6b}$, an open source large language model as our base to construct *SocraticLM*, because it is easy to fine-tune and has not been pre-trained with teaching capabilities, which can better verify the effect of our *SocraTeach* dataset. In the future, we will use our dataset to fine-tune more LLMs and explore their potential for teaching and intelligent education.

