# OpenReview forum: "SocraticLM: Exploring Socratic Personalized Teaching with Large Language Models"
_NeurIPS.cc/2024/Conference — NeurIPS 2024 spotlight_

### Official Review · Reviewer_3PwF · 2024-07-12

**Soundness:** 3
**Presentation:** 3
**Contribution:** 4
**Rating:** 8
**Confidence:** 5

**Summary:**

This paper aims to introduce a “Thought-provoking” paradigm into LLM-based personal teaching. The authors propose an innovative “Dean-Teacher-Student” pipeline with three LLM-based agents to collect Socratic teaching data. During this process, the authors also contribute a student cognitive system to simulate six types of authentic students with the “Student” agent. Then, the authors adopt data augmentation for four crucial teaching abilities and collect more single-round dialogue data. Finally, the authors investigate three training strategies to balance the problem-solving ability and Socratic teaching ability of LLMs. The fine-tuned SocraticLM, shows great teaching performance compared with several LLMs including GPT4 and EduChat. The authors also provide sufficient experiments to validate the importance of single-round data, training strategies, which makes the method self-contained and instructive.

**Strengths:**

- Motivation: I agree that the idea of LLMs-based teaching is essential for current intelligent education systems, and the existing methods are hard to achieve satisfactory teaching effects through simply answering students’ questions. Therefore, this paper is beneficial for practical applications.

- Dataset Contribution: I think one of the most important contributions of this paper is the proposed SocraTeach dataset that consists of high-quality teaching dialogues. It is the first public large-scale Socratic teaching dataset. The authors claim to release it, which can benefit the community to conduct more research for teaching-LLM.

- Methodology: The proposed “Dean-Teacher-Student” is reasonable and easy to reproduce. The introduction of student cognitive system that simulates six kinds of students guarantee the diversity of the teaching dialogue data. It has potentials to be generalized to other works.

- Comprehensive Assessment: Sufficient experiments is conducted to demonstrate the performance of SocraticLM over several LLMs, which makes the improvement convincing. Moreover, I think the analyses with different data scale is beneficial for reproducing and assist for building other educational LLMs.

**Weaknesses:**

Some issues that hope to be further addressed by authors:
- It is rational and acceptable to use GPT4 to construct data, but I am interesting about whether the performance of SocraticLM will be limited by GPT4 (please refer to question 1 below).
- I recommend the authors to provide more explanations of some experimental results, which can better reflect the effectiveness of the proposed SocraticLM (please refer to question 2 below).

**Questions:**

- In my opinion, the quality of the collected SocraTeach dataset highly depends on the Teacher agent, which is implemented with GPT4. Therefore, if GPT-4 itself cannot correctly solve a problem, can it still serve as a teacher in an educational role?
- I notice that in Table 2, SocraticLM performs even better in problem-solving accuracy on MAWPS dataset. Could the authors provide some explanations for this?

**Limitations:**

Please see questions above.

---

> ### Author Rebuttal · Authors · 2024-08-02
>
> We sincerely appreciate your affirmation of our motivation, the contribution of our dataset, the novelty and generalization ability of our pipeline, and our sufficient experiments and comprehensive assessment.
>
> $\bf{Q1}$:If GPT-4 itself cannot correctly solve a problem, can it still serve as a teacher in an educational role?
>
> $\bf{A1}$:Thank you very much for your insightful question. We believe it still can serve as a teacher for two main reasons. Firstly, the Dean agent can judge and correct each round of the Teacher agent's instruction. Within a single round, the instruction needed to be judged usually involves just one reasoning step, making this process easier than having GPT-4 solve a problem directly. Therefore, even if GPT-4's problem-solving ability is limited, it still has the potential to judge and revise the instruction accurately. Secondly, our pipeline does not rely on GPT-4's inherent problem-solving ability because we input the correct solutions into the prompts for both the Dean and Teacher agents. In other words, our pipeline is indeed having SocraticLM teach the correct solution to students rather than solving the problem itself. Therefore, the problem-solving ability of GPT-4 does not limit the effectiveness of our pipeline. Your question is very thought-provoking, and we will supplement these discussions in the revised version of our paper.
>
> $\bf{Q2}$:Why SocraticLM performs even better in problem-solving accuracy on MAWPS dataset?
>
> $\bf{A2}$:Thanks for your valuable question. We think the reason is that from fine-tuning on our SocraTeach dataset, SocraticLM indeed learns to answer multiple questions from students about various aspects of a single problem (e.g., asking about each reasoning step and the involved knowledge). This process may allow SocraticLM to develop a deeper understanding of the problem-solving process, which in turn can improve its problem-solving accuracy.

---

> > ### Comment · Reviewer_3PwF · 2024-08-08
> >
> > Thanks for your answer. These responses fulfill my doubts and reinforce my score.

---

> > > ### Author Response · Authors · 2024-08-08
> > >
> > > Many thanks for your quick response! We are very happy to resolve your concerns and will add all suggested modifications into the revised version based on your comments. Thank you again for the time you took to review our paper and your affirmation of our work. If there are any further questions, please feel free to raise them, and we can discuss any questions at any time. We will try our best and respond as soon as possible.

---

### Official Review · Reviewer_sW5a · 2024-07-13

**Soundness:** 3
**Presentation:** 3
**Contribution:** 4
**Rating:** 7
**Confidence:** 4

**Summary:**

The authors propose a novel method based on Socratic teaching for improving LLM teaching abilities.

**Strengths:**

- novel, interesting, and well-described method for improving LLM teaching ability
- creation and release of a useful, novel teaching dialogue dataset
- propose and validate novel ways of testing LLM teaching ability
- extensive experiments; I especially appreciate the ablation study
- code released
- I really appreciate the figures which do a great job of laying out the proposed methods and providing examples

**Weaknesses:**

- results could benefit from having standard errors or confidence intervals (authors say 'Yes' to this in NeurIPS checklist but only list having the kappa score in justification)
- there seem to be a large number of related methods (e.g., search for "socratic LLM teaching" on Google Scholar), the authors should contextualize their work among this literature in the related works section, and clarify how their method differs from and outperforms existing methods from that literature (e.g., a goog example is Socratic Playground - https://arxiv.org/abs/2406.13919)

Minor (did not affect score):
- Line 122: "While some student" --> "While some students"
- Line 173: "such not providing" --> typo here
- Line 312: "refuse" --> "refuses" or "refused"
- Table 1 caption: "denote the" --> "denotes"
- Line 595: "Border" -> "Broader"

Ethics flag:
- Research involved human subjects but unclear if authors had IRB approval. Not sure whether IRB approval is needed or not in this case so flagging for review by relevant experts just in case (note: the authors say the paper involves no crowdsourcing or human subjects in question 15 of the NeurIPS checklist, right after saying it does in question 14).

**Questions:**

- why does performance fall on some metrics at 125% data scale?
- will the full SocraTeach dataset be publicly released?
- I would be open to increasing my score if concerns from the previous section are addressed

**Limitations:**

- authors discuss limitations but would be worth it to add that current experiments are only in English, and that current human evaluations come from a very small (and likely not broadly representative) sample

---

> ### Comment · Area_Chair_WMJU · 2024-07-31
>
> Hello. This submission has been labeled by the reviewer for additional ethics review, I recommond the authors to priortize their response to the ethics concern so the ethics reviewers could get a clearer picture of the situation as early as possible. Please find above the ethics concerns raised by Reviewer sW5a.
>
> The purpose of the additional ethics review is to help the authors mitigating potential ethical, legal or societal harm at an early stage (if there's any). Thank you for your understanding.

---

> ### Author Rebuttal · Authors · 2024-08-02
>
> We sincerely appreciate your affirmation of the novelty of our SocraticLM, the value of our constructed teaching dialogue dataset, the innovation of our teaching ability evaluation system, and the clarity of our writing.
>
> For your concerns regarding the ethics flag, we want to express our sincere gratitude for your attention and sorry for the misunderstanding. In this paper, we aim to construct a SocraticLM for personalized teaching. To achieve this, we first propose a “Dean-Teacher-Student” (DTS) pipeline to collect large-scale Socratic teaching dialogue data, where we implement all agents with GPT-4 and do not incorporate human subjects. As for our evaluation system, similar to other works, we invited 10 annotators to assess and compare the teaching instructions of different LLMs to test their performances. As shown in our annotation template in Appendix F, this process also does not involve research on humans or privacy/security risks. Following your suggestion, we will provide the necessary clarification and corrections to the conference and give a clearer introduction of our annotation process in the revised version. The following are the responses to your other questions:
>
> $\bf{Q1}$:Standard errors or confidence intervals.
>
> $\bf{A1}$:Thanks for your constructive comments. We add the following results:
> Model|Overall|IARA|CARA|SER|SRR
>  -|-|-|-|-|-
> ChatGPT|0.29±0.024|0.42±0.034|0.93±0.001|0.62±0.052|0.19±0.020
> GPT4|0.50±0.000|0.76±0.015|0.91±0.017|0.65±0.060|0.55±0.050
> ChatGLM3|0.11±0.006|0.18±0.014|0.87±0.022|0.46±0.015|0.07±0.006
> SocraticLM|0.62±0.029|0.83±0.028|0.98±0.015|0.74±0.039|0.78±0.026
>
> $\bf{Q2}$:How this paper differs from and outperforms existing methods.
>
> $\bf{A2}$:Thanks for your thought-provoking question. As summarized in our related work section, existing methods on Socratic teaching with LLMs can be divided into two categories. The first uses a general LLM (e.g., ChatGPT, GPT-4) to assist in conversation design in courses[1], content authoring[2], explaining learning-paths[3], and providing feedback[4]. Notably, the work you mentioned[5] uses GPT-4 to create an innovative Socratic Playground for Learning, constructing diverse learning scenarios where GPT-4 interacts with students in a Socratic manner. The second category collects data to train a specialized teaching LLM, with EduChat[6] being the most representative (also used as our baseline).
>
> Compared to them, our SocraticLM improves from two main aspects. 1) When constructing the SocraTeach dataset, we build a student cognitive system to simulate six kinds of authentic students and enhance four key teaching abilities. This enables SocraticLM to handle more complex teaching scenarios and have more comprehensive and systematic teaching abilities. 2) Our DTS pipeline contains a novel Dean agent to judge and revise the GPT-4-based Teacher agent, addressing the limitations that existing general LLMs may still make for a bad teacher[7] and improving teaching quality. Experiments show that SocraticLM outperforms GPT-4 and EduChat across various dimensions, verifying the effectiveness of our pipeline and model.
>
> Thank you for pointing out the excellent works and we will cite them in our revised paper.
>
> [1] Chatgpt in the generalized intelligent framework for tutoring.
>
> [2] Ruffle&riley: Towards the automated induction of conversational tutoring systems.
>
> [3] Supporting student decisions on learning recommendations: An llmbased chatbot with knowledge graph contextualization for conversational explainability and mentoring.
>
> [4] How can i get it right? using gpt to rephrase incorrect trainee responses.
>
> [5] SPL: A Socratic Playground for Learning Powered by Large Language Model.
>
> [6] Educhat: A large-scale language model-based chatbot system for intelligent education.
>
> [7] The ai teacher test: Measuring the pedagogical ability of blender and gpt-3 in educational dialogues.
>
> $\bf{Q3}$:Minor typos.
>
> $\bf{A3}$:Thank you very much for thoroughly reading our paper and pointing out them. We will revise our paper carefully.
>
> $\bf{Q4}$:Performance fall on some metrics at 125% data scale?
>
> $\bf{A4}$:Many thanks for your insightful question. In Figure 4, at the 125% data scale, the metrics that decline are IARA and overall quality, indicating that the root cause is a decrease in the model's ability to identify incorrect answers (the decline in overall quality is a subsequent result). This may be because, with the increase in multi-round dialogue data, the proportion of single-round dialogue data for “Incorrect reply” decreases. When multi-round data scale exceeds 125%, the proportion of this single-round dialogue data may fall below a certain threshold, which results in diminishing effectiveness. Many thanks for pointing out this phenomenon. We will supplement these discussions and explanations in the revised paper.
>
> $\bf{Q5}$:Will dataset be released?
>
> $\bf{A5}$:Yes! We have already made the test set of our SocraTeach dataset and the model training code available through an anonymous repository (https://anonymous.4open.science/r/NeurIPS-4310). If our paper is accepted, we will release the full dataset as soon as possible.
>
> $\bf{Q6}$:Current experiments are only in English and human evaluations come from a small sample.
>
> $\bf{A6}$:Thanks for your constructive comments. Although our current dataset is in English, as mentioned in Section 3.2, our "Dean-Teacher-Student" pipeline is general and can easily be applied to other datasets, such as the Chinese Math23K dataset. For human evaluations, we sampled over 1,000 dialogues from SocraTeach dataset to calculate metrics and calculated the Kappa score within the annotators. The results demonstrate the consistency of the manual evaluations and the effectiveness of our SocraticLM. Inspired by your comments, we will consider constructing multilingual teaching datasets and training judging networks based on human-annotated results for larger-scale automatic testing in the future.

---

> > ### Author Response · Authors · 2024-08-13
> >
> > We wish to once again express our great appreciation for the time you have taken to review our paper. We would appreciate your feedback on whether your main concerns have been adequately addressed. We truly value your understanding and support, and will carefully revise the paper according to your suggestions. Thank you very much!

---

> > ### Comment · Reviewer_sW5a · 2024-08-13
> >
> > Thank you for the rebuttal! My concerns have been addressed and I have increased my score accordingly.

---

> ### Author Response · Authors · 2024-08-14
>
> Many thanks for your response and increasing your score! We are very happy to address your concerns and will add all suggested clarifications and modifications into the revised version following your comments. Thank you again for the time you took to review our paper and your affirmation of our work. If there are any further questions, please feel free to raise them, and we can discuss any questions at any time. We will try our best and respond as soon as possible.

---

### Official Review · Reviewer_3yrk · 2024-07-13

**Soundness:** 3
**Presentation:** 3
**Contribution:** 3
**Rating:** 8
**Confidence:** 4

**Summary:**

In this paper, the authors propose a novel SocraticLM to address the limitations of existing personal teaching methods that follow a “Question-answering” paradigm. To do this, the authors first propose a novel “Dean-Teacher-Student” pipeline to collect multi-round Socratic teaching dialogues, where the authors also design a student cognitive system to guides the agent behaviors. Then, the authors elaborate on four teaching abilities and propose corresponding data augmentation. Next, the authors fine-tune ChatGLM3 on the collected dataset with three strategies to balance teaching and reasoning abilities. Finally, the authors design the first systematic evaluation system for the teaching quality of LLMs. Experimental results clearly show the superiority of SocraticLM. Overall, this paper is easy to follow and presents a nice structure.

**Strengths:**

This paper has the following strengths:
1.The motivation to introduce a Socratic-teaching LLM that follows “Thought-Provoking” paradigm is reasonable and necessary for AI and intelligent education.
2.The “Dean-Teacher-Student” multi-agent pipeline can sufficiently collect high-quality teaching dialogues. Besides, this pipeline is general enough to expand to other domains. Additionally, the data augmentation for four teaching abilities makes sense and is clearly organized.
3.This paper discuss and design a comprehensive systematic evaluation system for the teaching quality of LLMs, which provides an effective way in the related research.
The experimental results compared with 8 LLMs are sufficient, and the significant improvement over GPT4 clearly verifies the effectiveness of SocraticLM. Besides, the experiments for evaluating the necessity of component of SocraticLM including single-round dialogues, training strategies, making the paper instructive.

**Weaknesses:**

I have some concerns:
1.One of the most important roles in the proposed pipeline is Dean agent, which plays a supervisory role compared to previous work. Thus, I recommend to add more analyses of its effectiveness in experiments. (please refer to question 1 below).
2.The single-round teaching dialogues are also important in this paper because they correspond to four crucial teaching abilities. Thus, I recommend to add more explanations of the experimental results in Section 6.2. (please refer to question 2 below).
3.I found some typos, for example,
--Line 193, “a student need” should be “a student needs”
--Line 275, “dialogue” should be “dialogues”

**Questions:**

1.I think the Dean agent is important in the proposed pipeline. Therefore, I have the question: How to assess the importance of Dean agent? Can it be reflected by comparing the results of SocraticLM and GPT4?
2.In Table 1, comparing the results of “w/o DiaS” and “w/o Correct”, I have a minor question: why the CARA metric of “w/o Correct” is lower than “w/o DiaS”? In my opinion, “w/o Correct” incorporates more single-round teaching dialogues, and thus its performance should be better.

---

> ### Author Rebuttal · Authors · 2024-08-02
>
> We sincerely appreciate your affirmation of the significance of our motivation, the innovation and generalizability of our pipeline, the contributions of our evaluation system, and the effectiveness of our SocraticLM.
>
> $\bf{Q1}$:How to assess the importance of Dean agent?
>
> $\bf{A1}$:Many thanks for your careful comment. Yes, as you have noted, the effectiveness of Dean agent can be reflected by comparing the performance of SocraticLM and GPT-4. For a given preceding dialogue, the Dean agent judges and corrects the responses generated by the GPT-4 simulated Teacher agent. This is the fundamental root for the differences between GPT-4 and SocraticLM after fine-tuning. Therefore, the superior performance of SocraticLM compared to GPT-4 as shown in Table 1 can directly demonstrates that Dean agent is effective and necessary.
>
> $\bf{Q2}$:Why the CARA metric of “w/o Correct” is lower than “w/o DiaS”
>
> $\bf{A2}$:Thank you very much for the question. We think the reason is that the "w/o Correct" model was still fine-tuned on 10K single-round dialogues corresponding to students’ "Incorrect-reply" as explained in Section 3.4. This led the model to develop a stronger tendency to perceive a student's reply as incorrect, causing it to incorrectly classify some correct student replies as wrong. As a result, the model's performance on the CARA metric was worse compared to the model that did not include single-round data training (i.e., “w/o DiaS”). This phenomenon indicates that it is necessary to construct data for the "Correct-reply" category and balance it with the "Incorrect-reply" category.
>
> $\bf{Q3}$:Exist some typos.
>
> $\bf{A3}$:Thank you for pointing this out. We will carefully review and polish the writing of our paper.

---

> > ### Author Response · Authors · 2024-08-13
> >
> > We wish to once again express our great appreciation for the time you have taken to review our paper. We would appreciate your feedback on whether your main concerns have been adequately addressed. We truly value your understanding and support, and will carefully revise the paper according to your suggestions. Thank you very much!

---

> > ### Comment · Reviewer_3yrk · 2024-08-13
> > **comment from reviewer**
> >
> > Thank you for clarifying the doubts. All my concerns have been addressed.

---

> > > ### Author Response · Authors · 2024-08-14
> > >
> > > Many thanks for your response! We are very happy to address your concerns and will add all your suggested modifications into the revised version. Thank you again for the time you took to review our paper and your affirmation of our work. If there are any further questions, please feel free to raise them, and we can discuss any questions at any time. We will try our best and respond as soon as possible.

---

### Official Review · Reviewer_twUm · 2024-07-23

**Soundness:** 3
**Presentation:** 2
**Contribution:** 3
**Rating:** 6
**Confidence:** 3

**Summary:**

In this paper, the authors fine-tuned a language model on synthetic data for Socratic Personalized Teaching and evaluated the performance of the proposed model in comparison with a number of baseline options.

The authors proposed a multi-agent data synthesis pipeline. Using GPT-4, the authors simulated responses from both the AI teacher (the language model) and the student (the human user.) The author also included a "dean" agent role for evaluating proposed responses against a set of principles for Socratic Personalized Teaching. The authors fine-tuned a language model on the resultant synthetic data. The authors recruited human annotators to evaluate the overall quality of the fine-tuned model. Based on their synthetic dataset, the authors proposed a set of additional metrics for evaluating the performance of language models in the context of Socratic Personalized Teaching.

The authors also shared insights on ways to prevent forgetting during the fine-tuning process.

**Strengths:**

The multi-agent pipeline for synthesizing data in the field of Socratic Personalized Teaching is a rather novel approach. The proposed method of leveraging synthetic data for evaluation resolves the limitation where human-written data in this field is difficult to obtain.

Results on the impact of the "Scale of Problem-solving Data" (Section 6.3) provides useful insights into the trade-off between fine-tuning data (teaching style) and data from the original domain (ability to solve problems.)

Examples listed in the appendix suggest that the proposed approach indeed performs better than general-purpose baseline models such as GPT-4 or GPT-3.5.

**Weaknesses:**

Given that the evaluation method is a key contribution of this paper, the authors might also want describe how their approach compare with related works in evaluating LLMs for education.

Instead of specifying one of the six "Student Cognitive State" when generating a response as a student, the authors allowed the LLM to pick one of these options on its own (see section B.1,) resulting in an imbalanced data distribution (see figure 6 (b) in the appendix.) It might be helpful if the authors can elaborate more on this design choice, as personally I find it unclear how this imbalanced distribution might affect the model performance in the real world.

It is nice to see that fine-tuning ChatGLM3-6B on the training split of the synthetic dataset improved performance on the test split, exceeding that of GPT-4, which generated the synthetic responses. Nevertheless, it is unclear from the paper how robust the model would be when the data distribution changes. One possible approach might be to collect additional MOOC question-answer data (as the authors already did) and test the model on these examples.

**Questions:**

It is unclear whether the seed questions from GSM8K and MAWPS behind the synthetic data for evaluation are entirely separate from the ones for generating the train split of the synthetic dataset. For example, would the same question from GSM8K appear in both the synthetic evaluation set and the synthetic fine-tuning set? Line 610 in the limitation section seems to allude to that, but wasn't entirely clear in my opinion.

While "Overall Quality" is evaluated through human annotation (see appendix F), what about the metrics IARA, CARA, SER, and SRR? Are these human-in-the-loop as well? It will be helpful if you can elaborate on these details.

As a side note, the repository (https://anonymous.4open.science/r/NeurIPS-4310) does not seem to contain code for data synthesis.

**Limitations:**

The use of ChatGLM3-6B as the base model is a possible limitation- given that the base model correctly solves only around 65% of the GSM8K questions (vs ~90% for GPT-4,) it might be unclear whether the model would be able to consistently give a correct answer. The authors acknowledged this limitation in Appendix J by stating that the same synthetic dataset can be used to fine-tune other LLMs. This limitation (related to forgetting during fine-tuning) was also touched upon in section 6.3.

Overall, I find the handling of this limitation quite reasonable- given that this paper is about Socratic Personalized Education, not about improving scores on GSM8K. Nevertheless, the authors might want to adjust the evaluation setup to account for this limitation- e.g., excluding from the evaluation GSM8K questions that cannot be solved using the original ChatGLM3-6B.

---

> ### Author Rebuttal · Authors · 2024-08-02
>
> We sincerely appreciate your affirmation of the novelty of our pipeline, the contribution of our evaluation system, and the significance of our experiments.
>
> $\bf{Q1}$:Compare with related works in evaluating LLMs for education?
>
> $\bf{A1}$:Thanks for your valuable question. The related works can be divided into two categories. The first involves objective similarity assessments with human-annotated instruction, using metrics like BLEU and BERTScore[1]. The second involves subjective human evaluations, such as calculating the percentage where student reaches the correct answer given the instructions[1], analyzing the correlation between student grade changes and the use of LLMs[2,3], and issuing questionnaires[4].
>
> Compared to them, our evaluation system offers several advantages:
>
> 1. More comprehensive and adequate assessment. It includes evaluations of overall teaching quality and four key teaching abilities, which current works lack a systematic organization.
>
> 2. Better comparability. Since a real student can only be taught by one LLM at a time, it is hard to compare the teaching effects of different LLMs by the aforementioned subjective human evaluations. In comparison, our evaluation is based on shared teaching dialogues, which can support the fair evaluation of multiple LLMs simultaneously.
>
> 3. More extensive. The latest dataset[1] contains only 600 testing dialogue data, which is far less than our data volume.
>
> Frollowing your suggestions, we will supplement these discussions in our paper to highlight our contributions.
>
> [1] A dialogue tutoring dataset with rich pedagogical properties grounded in math reasoning problems.
>
> [2] Gptutor: a chatgpt-powered programming tool for code explanation.
>
> [3] GPT-Empowered Personalized eLearning System for Programming Languages.
>
> [4] Recipe: How to integrate chatgpt into efl writing education.
>
> $\bf{Q2}$:Why allow the LLM to pick instead of specifying student cognitive states? The imbalanced distribution affects the performance?
>
> $\bf{A2}$:Thanks for your insightful question. We allow the LLM to pick a student cognitive state for two reasons. First, real teaching processes are inherently complex, which may cover multiple cognitive dimensions (e.g., calculation, knowledge mastery) within a single teaching dialogue. Thus, manually specifying the state may limit the diversity and hinder the simulation of real teaching process. Second, it may disrupt the dialogue fluency. For example, if the Teacher agent asks "Do you know the concept of trigonometry?", it would be unnatural to ask the Student agent to response according to a weak "Problem Understanding" state, which may lead to an almost irrelevant reply, making it less like a real teacher-student conversation.
>
> The imbalanced distribution will affect the model's performance. For instance, for a real student with weak "Instruction Understanding" state (i.e., type 2 in Figure 6), a model trained on an imbalanced dataset with much less data on type 2 may struggle to provide appropriate instructions. To address this, in Section 3.4, we employ data augmentation for four key teaching abilities, and the ablation study in Section 6.2 clearly verifies the necessity of this enhancement.
>
> $\bf{Q3}$:Robustness when the data distribution changes (e.g., test on more MOOC question-answer pairs) and what about the dataset partition?
>
> $\bf{A3}$:Thanks for your meticulous attention to our data collection. Firstly, sorry for our unclear statements. In Section 3.4, we collected 200 genuine student inquiries from MOOCs that are unrelated to teaching to conduct data augmentation for students’ “Irrelevant” responses, rather than collecting question-answer pairs. Secondly, we greatly appreciate your concern regarding the model robustness. As stated in lines 316-317 of Section 6, we remove the dialogues for questions in the evaluation set during training. Therefore, the issue of "the same questions appear simultaneously in both the evaluation set and the fine-tuning set" does not arise. This also addresses your point about the distribution shift or the need to test on new MOOC data, as our test set has not appeared in training, which validates the robustness of our model.
>
> $\bf{Q4}$:Elaborate on metrics IARA, CARA, SER, and SRR.
>
> $\bf{A4}$:Yes! These metrics are also evaluated manually. Since they are considered objective binary classification tasks as explained in Section 5, when annotating, we simply provide the preceding dialogue and ask annotators to evaluate whether the LLM's response "identifies the incorrect/correct student reply" (for IARA/CARA), "addresses the student's question" (for SER), or "refuses to answer the student's irrelevant question" (for SRR).
>
> $\bf{Q5}$:Code for data synthesis.
>
> $\bf{A5}$:Thanks for your comments. We supplement all the code of our agents needed for data synthesis in the folder “dataset_synthesis” in our repository. If the paper is accepted, we will also release all the data and code for public use as soon as possible.
>
> $\bf{Q6}$:Evaluation that excludes the GSM8K questions that ChatGLM3 cannot be solved.
>
> $\bf{A6}$:Thanks for your constructive suggestion. From the Table below, we observe that the performances of LLMs improve, but the increase is not particularly significant. We think this is because, in constructing our SocraTeach dataset, the prompts for Dean and Teacher agents include the correct solutions for the problems. In other words, we do not require ChatGLM3 to solve the problems directly but rather to learn how to teach the correct solution to a student. Thus, the problem-solving ability of ChatGLM3 does not set the upper limit for the teaching ability of our SocraticLM.
> Model|Overall|IARA|CARA|SER|SRR
>  -|-|-|-|-|-
> ChatGPT|0.34±0.079|0.50±0.010|0.95±0.004|0.68±0.079|0.23±0.001
> GPT4|0.50±0.000|0.78±0.059|0.91±0.014|0.68±0.002|0.53±0.005
> ChatGLM3|0.14±0.039|0.21±0.052|0.88±0.010|0.49±0.079|0.07±0.009
> SocraticLM|0.66±0.045|0.84±0.091|1.00±0.000|0.78±0.076|0.77±0.006

---

> > ### Author Response · Authors · 2024-08-13
> >
> > We wish to once again express our great appreciation for the time you have taken to review our paper. We would appreciate your feedback on whether your main concerns have been adequately addressed. We truly value your understanding and support, and will carefully revise the paper according to your suggestions. Thank you very much!

---

### Author Rebuttal · Authors · 2024-08-02

We sincerely thank all reviewers’ efforts in reviewing our paper. We would like to thank all of them for providing constructive and valuable feedback, which we will leverage to improve this work. We are encouraged by the positive comments from reviewers, including:
- **Motivation**: “The motivation to introduce a Socratic-teaching LLM that follows “Thought-Provoking” paradigm is reasonable and necessary for AI and intelligent education” (Reviewer 3yrk), “LLMs-based teaching is essential for current intelligent education systems” (Reviewer 3PwF).

- **Method**: “novel multi-agent pipeline” (Reviewer twUm, Reviewer 3yrk), “shared insights on ways to prevent forgetting” (Reviewer twUm), “novel SocraticLM” (Reviewer 3yrk), “interesting, and well-described method” (Reviewer sW5a), “useful, novel teaching dialogue dataset” (Reviewer sW5a), “novel ways of testing LLM teaching ability” (Reviewer sW5a), “innovative, reasonable and easy to reproduce” (Reviewer 3PwF), “self-contained and instructive” (Reviewer 3PwF).

- **Experimental Results**: “provides useful insights” (Reviewer twUm), “performs better than baseline models such as GPT-4, GPT-3.5, and EduChat” (Reviewer twUm, Reviewer 3PwF), “sufficient experimental results” (Reviewer 3yrk, Reviewer 3PwF), “significant improvement over GPT4” (Reviewer 3yrk), “especially appreciate the ablation study” (Reviewer sW5a).

- **Significance**: “resolves the limitation where human-written data in this field is difficult to obtain” (Reviewer twUm), “evaluation system provides an effective way in the related research.” (Reviewer 3yrk), “contribute a student cognitive system” (Reviewer 3PwF), “the first public large-scale Socratic teaching dataset” (Reviewer 3PwF), “beneficial for practical applications” (Reviewer 3PwF), “beneficial for reproducing and assist for building other educational LLMs.” (Reviewer 3PwF), “code and dataset released” (Reviewer sW5a, Reviewer 3PwF).

**[Response to Ethics Reviewers]**

We deeply appreciate your thoughtful consideration of the ethics flag of our paper. Firstly, our paper focuses on introducing a Socratic-style teaching LLM, SocraticLM. To achieve this, we propose a “Dean-Teacher-Student” pipeline to first collect a teaching dialogue dataset, SocraTeach. As we introduce in Section 3, the entire pipeline is implemented without human involvement, as all three agents in the pipeline are simulated using GPT-4. Therefore, from a technical perspective, our work does not raise ethical issues. Secondly, regarding our evaluation system, similar to other works that involve human evaluation of LLM outputs, we invited 10 annotators to anonymously assess and compare the outputs (i.e., teaching instructions) of different LLMs for performance testing. As shown in the annotation template in Appendix F, this process does not involve testing of annotators or research with human subjects. While our paper aimed to present and evaluate a novel personalized teaching LLM, your feedback highlights the necessity of addressing the potential ethical pitfalls. Many thanks for your concerns, and we will revise the paper to incorporate a dedicated subsection that discusses and clarifies our annotation process.

---

### Decision · Program_Chairs · 2024-09-25

**Decision:**

Accept (spotlight)

**Comment:**

In this paper, the authors propose a Socratic "Thought-Provoking'' teaching paradigm that simulates the role of a real classroom teacher in actively engaging students in the thought process required for genuine problem-solving mastery. They design a "Dean-Teacher-Student'' multi-agent system for data collection, with a set of authentic teaching scenarios integrated. They show that an LM fine-tuned on such data (with some strategies balancing teaching and reasoning abilities) could outperform strong baselines such as GPT4 in its teaching performance.

All four reviwers agree on the good quality of this submission: the work is novel and sound; the problem is well-motivated; the proposed method is general enough so it can be relatively easily generalized to other domains/scenarios; the presentation is clear; the experimental design is good.

To me this is a clear accept, but to help the authors further improve their work, let me summarize the reviewers' suggestions:
- Discuss the connection between this work and prior work in the area of LLM evaluation in educational/pedagogical scenarios. (twUm, sW5a)
- Provide more analyses on how the Dean role is useful. (3yrk)
- Provide more analyses related to the single-round teaching dialogue setting, specifically, in Section 6.2. (3yrk)
- Add error bars. (sW5a)
- Add discussion about performance fall on some metrics at 125% data scale in Figure 4. (sW5a)
- Add discussion about how the pipeline could be robust in cases where the LLM Teacher and Dean are imperfect. (3PwF)